# Comprehensive at-arrival transcriptomic analysis of post-weaned beef cattle uncovers type I interferon and antiviral mechanisms associated with bovine respiratory disease mortality

Matthew A. Scott[1]*, Amelia R. Woolums[1], Cyprianna E. Swiderski[2], Andy D. Perkins[3], Bindu Nanduri[4], David R. Smith[1], Brandi B. Karisch[5], William B. Epperson[1], John R. Blanton[5]

1 Department of Pathobiology and Population Medicine, Mississippi State University, Mississippi State, MS, United States of America, 2 Department of Clinical Sciences, Mississippi State University, Mississippi State, MS, United States of America, 3 Department of Computer Science and Engineering, Mississippi State University, Mississippi State, MS, United States of America, 4 Department of Basic Sciences, Mississippi State University College of Veterinary Medicine, Mississippi State University, Mississippi State, MS, United States of America, 5 Department of Animal and Dairy Sciences, Mississippi State University, Mississippi State, MS, United States of America

* mas1052@msstate.edu

**Data Availability Statement:** The datasets supporting the conclusions of this article are

## Abstract

### Background

Despite decades of extensive research, bovine respiratory disease (BRD) remains the most devastating disease in beef cattle production. Establishing a clinical diagnosis often relies upon visual detection of non-specific signs, leading to low diagnostic accuracy. Thus, post-weaned beef cattle are often metaphylactically administered antimicrobials at facility arrival, which poses concerns regarding antimicrobial stewardship and resistance. Additionally, there is a lack of high-quality research that addresses the gene-by-environment interactions that underlie why some cattle that develop BRD die while others survive. Therefore, it is necessary to decipher the underlying host genomic factors associated with BRD mortality versus survival to help determine BRD risk and severity. Using transcriptomic analysis of at-arrival whole blood samples from cattle that died of BRD, as compared to those that developed signs of BRD but lived (n = 3 DEAD, n = 3 ALIVE), we identified differentially expressed genes (DEGs) and associated pathways in cattle that died of BRD. Additionally, we evaluated unmapped reads, which are often overlooked within transcriptomic experiments.

### Results

69 DEGs (FDR<0.10) were identified between ALIVE and DEAD cohorts. Several DEGs possess immunological and proinflammatory function and associations with TLR4 and IL6. Biological processes, pathways, and disease phenotype associations related to type-I interferon production and antiviral defense were enriched in DEAD cattle at arrival. Unmapped

publicly available in the Gene Expression Omnibus (GEO) repository, https://www.ncbi.nlm.nih.gov/geo, under the accession number GSE136176, and the Sequence Read Archive (SRA) under the accession number SRP219429. All remaining data generated or analyzed during this study are included in this published article and its additional information files.

**Funding:** This project was funded and supported by the Mississippi State University Department of Pathobiology and Population Medicine, Mississippi State University Department of Animal and Dairy Sciences, and the Mississippi Agricultural and Forestry Experiment Station. The funders had no role in study design, data collection and analysis, decision to publish, or preparation of the manuscript.

**Competing interests:** The authors have declared that no competing interests exist.

reads aligned primarily to various ungulate assemblies, but failed to align to viral assemblies.

## Conclusion

This study further revealed increased proinflammatory immunological mechanisms in cattle that develop BRD. DEGs upregulated in DEAD cattle were predominantly involved in innate immune pathways typically associated with antiviral defense, although no viral genes were identified within unmapped reads. Our findings provide genomic targets for further analysis in cattle at highest risk of BRD, suggesting that mechanisms related to type I interferons and antiviral defense may be indicative of viral respiratory disease at arrival and contribute to eventual BRD mortality.

## Introduction

Although extensively researched, bovine respiratory disease (BRD) continues to be the most significant disease in post-weaned beef cattle in North America. BRD is a multifactorial disease complex, with contributing causative factors including primary viral infection, bacterial colonization of the upper and lower respiratory tract, and stressful events related to abrupt weaning, co-mingling with recently transported cattle, and novel feeding or housing environments [1–5]. These factors result in host-pathogen interactions that are exceedingly complex and definitive diagnosis of the inciting etiological agent(s) is not usually made. BRD diagnosis will typically rely on non-specific clinical signs including elevated rectal temperature, depressed demeanor, increased respiratory rate and effort, and anorexia [5–8]. However, this clinically based diagnosis has been shown to lack sensitivity and specificity [9, 10]. Therefore, post-weaned beef cattle at high risk of developing BRD are often mass medicated with antimicrobials at facility arrival (i.e. antimicrobial metaphylaxis) [11–13]. With growing public concerns regarding the relationship between the use of metaphylaxis in beef cattle and antimicrobial resistance, there is a need to recognize cattle at increased risk of developing BRD in order to implement more targeted therapeutic regimens.

In order to identify new methods of accurate BRD diagnosis, our previous research contrasted the whole blood transcriptomes of cattle that naturally acquired or resisted BRD [14]. Specifically, we identified upregulation of inflammatory-mitigating molecules and pathways at arrival in cattle that failed to develop naturally occurring clinical BRD. This prior research did not examine differentially expressed genes or pathways that segregate with disease severity. Therefore, we hypothesize that whole blood transcriptome profiles of cattle at arrival can identify biological functions that influence BRD severity; specifically, functions which distinguish cattle that are likely to die versus cattle that survive.

In the present study, we analyzed the at-arrival whole blood transcriptomes of post-weaned beef cattle that developed BRD. Specifically, we compared at-arrival whole blood transcriptomes from cattle that naturally acquired BRD within the first 28 days following arrival, stratifying cattle into severity groups defined by BRD-associated mortality. Our objectives were to uncover differentially expressed genes and associated processes and pathways that segregate with cattle at highest risk of BRD-associated mortality and to determine if reads from diseased cattle align to pathogens that promote inflammatory processes. Given the lack of sensitivity in clinical screening for BRD and the need to improve understanding of gene-by-environment interactions involved in BRD, the identification of gene products whose expression correlates

with the risk of BRD-associated mortality would advance management and diagnostic strategies for improving the outcome of post-weaned beef cattle in high-risk situations.

## Results

### Differential gene expression analysis

Alignment of reads from the six biological replicates to the ARS-UCD1.2 bovine reference assembly identified 32,976 unique genes. Following filtering for low expression, 15,755 genes were used for differential expression analysis between DEAD and ALIVE groups. Multidimensional scaling (MDS; Fig 1), which depicts the similarity of expression profiles between each animal in the analysis, demonstrated clustering of the three DEAD cattle (red; IDs 33, 52, 76). This indicates that the expression patterns of DEAD cattle are highly similar and are distinguishable from the cattle within the ALIVE group (blue; IDs 51, 75, 85). In contrast, the gene expression patterns of the three animals within the ALIVE group are clearly more dissimilar than the expression patterns of cattle in the DEAD group.

The three animals within the DEAD cohort are highly similar in gene expression and more distinct than the three animals within the ALIVE cohort, with leading fold-change of approximately 2-fold between the furthest points within the DEAD cohort. One animal within the ALIVE cohort (S_51) is the most dissimilar animal in terms of gene expression. Points represent each sample and their transformed Euclidean distance in two dimensions, discerned as leading log2-fold change between the pairwise distances of the top 500 genes that best differentiate each animal.

A total of 69 genes were differentially expressed (FDR < 0.10) between DEAD and ALIVE groups; 37 genes were upregulated and 32 downregulated in DEAD cattle compared to ALIVE cattle. A heatmap was generated from these 69 DEGs using z-scores calculated from Trimmed Mean of M-values (TMM) normalized counts (Fig 2). The resulting hierarchical clustering of DEG expression patterns for each individual segregates the six individuals into two groups according to their respective ALIVE and DEAD status and also provided a dendrogram of expression similarities. The complete list of DEGs with accompanying statistics are provided in S1 File.

Heatmap depicting gene expression directionality and hierarchical clustering of DEGs in each sample. Red or blue color intensities, respectively, correspond to increasing or decreasing gene expression. Dendrograms in the rows identify gene expression clusters. Note that

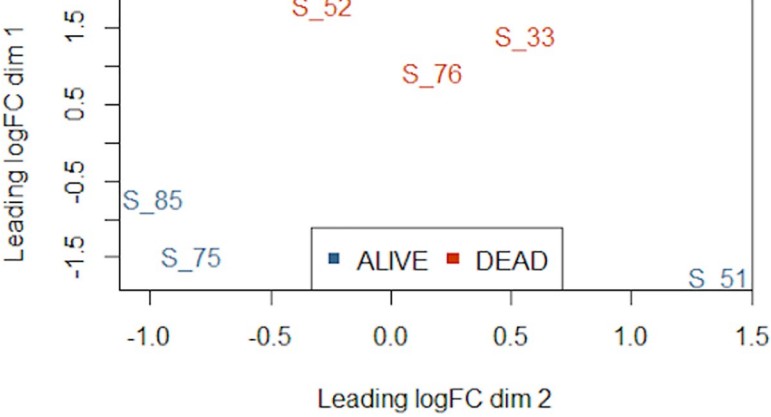

**Fig 1. Multidimensional scaling of RNA expression at arrival identifies expression clustering in cattle that die of BRD.**

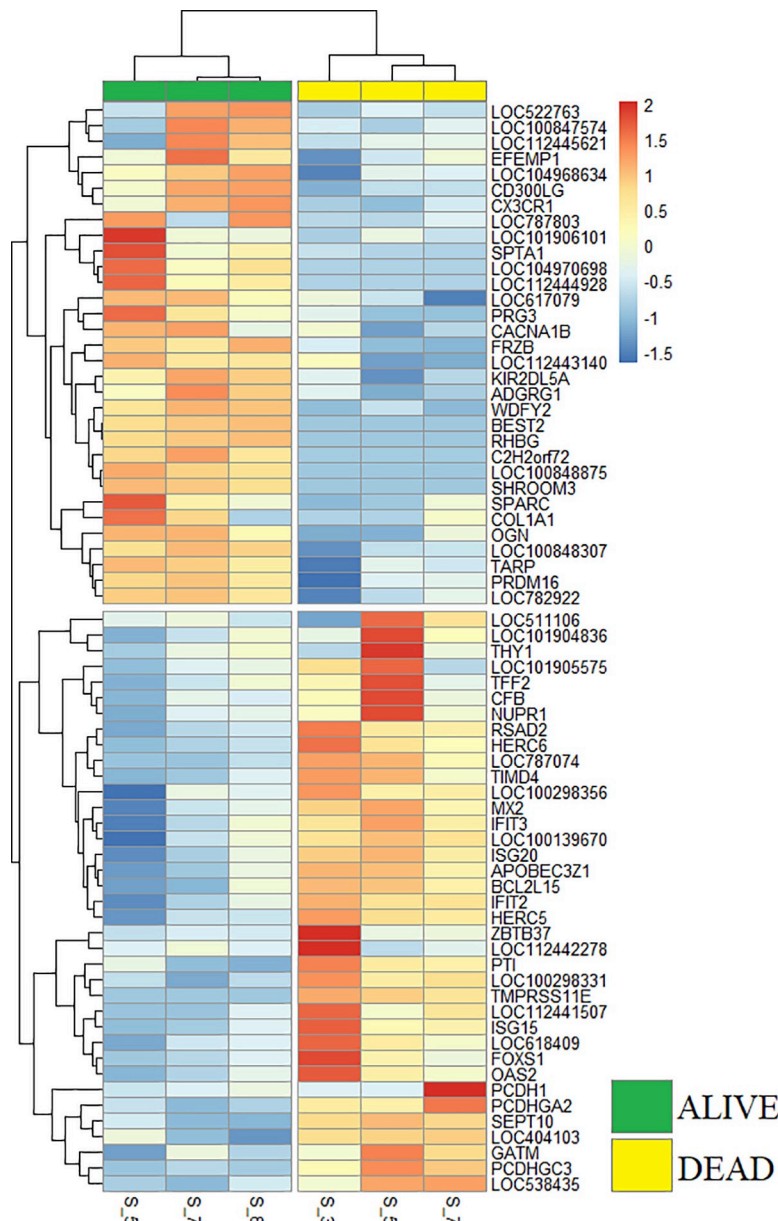

**Fig 2. Hierarchical clustering of gene expression from arrival blood separates ALIVE and DEAD cohorts.**

clustering of samples (columns) based on gene expression similarity segregates the samples into ALIVE and DEAD cohorts.

## Gene ontology, pathway, and disease phenotype enrichment

Gene ontology (GO) term enrichment of DEGs identified 34 significantly overrepresented biological processes (FDR < 0.05; Table 1). The top biological processes were primarily related to type I interferon signaling and response, viral defense mechanisms, and innate immune regulation involving cytokine signaling. These biological processes are chiefly composed of genes with higher expression in DEAD cattle, particularly those of the IFIT, ISG, HERC, and OAS

**Table 1. Top enriched GO-BP terms (FDR < 0.05).**

| Gene Set | Description | Size | P Value | FDR |
| --- | --- | --- | --- | --- |
| GO:0060337 | Type I interferon signaling pathway | 84 | 0 | 0 |
| GO:0071357 | Cellular response to type I interferon | 84 | 0 | 0 |
| GO:0034340 | Response to type I interferon | 89 | 1.11E-16 | 4.81E-13 |
| GO:0051607 | Defense response to virus | 235 | 2.70E-12 | 8.76E-09 |
| GO:0009615 | Response to virus | 319 | 5.55E-11 | 1.44E-07 |
| GO:0098542 | Defense response to other organism | 473 | 1.19E-10 | 2.58E-07 |
| GO:0019221 | Cytokine-mediated signaling pathway | 705 | 4.88E-10 | 9.06E-07 |
| GO:0045087 | Innate immune response | 827 | 3.00E-09 | 4.88E-06 |
| GO:0043207 | Response to external biotic stimulus | 899 | 7.70E-09 | 1.00E-05 |
| GO:0051707 | Response to other organism | 897 | 7.51E-09 | 1.00E-05 |
| GO:0002252 | Immune effector process | 1141 | 9.64E-09 | 1.14E-05 |
| GO:0009607 | Response to biotic stimulus | 926 | 1.07E-08 | 1.16E-05 |
| GO:0045071 | Negative regulation of viral genome replication | 50 | 1.85E-08 | 1.85E-05 |
| GO:0035455 | Response to interferon-alpha | 20 | 3.03E-08 | 2.63E-05 |
| GO:0071345 | Cellular response to cytokine stimulus | 1015 | 2.99E-08 | 2.63E-05 |
| GO:0034097 | Response to cytokine | 1100 | 7.29E-08 | 5.92E-05 |
| GO:1903901 | Negative regulation of viral life cycle | 74 | 1.37E-07 | 1.04E-04 |
| GO:0006952 | Defense response | 1518 | 2.86E-07 | 2.07E-04 |
| GO:0045069 | Regulation of viral genome replication | 87 | 3.09E-07 | 2.11E-04 |
| GO:0048525 | Negative regulation of viral process | 88 | 3.27E-07 | 2.12E-04 |
| GO:0006955 | Immune response | 1919 | 5.73E-07 | 3.55E-04 |
| GO:0019079 | Viral genome replication | 114 | 1.19E-06 | 7.02E-04 |
| GO:1903900 | Regulation of viral life cycle | 135 | 2.74E-06 | 1.55E-03 |
| GO:0043901 | Negative regulation of multi-organism process | 165 | 7.32E-06 | 3.96E-03 |
| GO:0002376 | Immune system process | 2778 | 8.67E-06 | 4.35E-03 |
| GO:0050792 | Regulation of viral process | 171 | 8.71E-06 | 4.35E-03 |
| GO:0043903 | Regulation of symbiosis, encompassing mutualism through parasitism | 198 | 1.77E-05 | 8.52E-03 |
| GO:0035456 | Response to interferon-beta | 33 | 2.24E-05 | 1.04E-02 |
| GO:0043900 | Regulation of multi-organism process | 367 | 2.74E-05 | 1.23E-02 |
| GO:0009605 | Response to external stimulus | 2282 | 2.87E-05 | 1.25E-02 |
| GO:0002697 | Regulation of immune effector process | 381 | 3.38E-05 | 1.41E-02 |
| GO:0032020 | ISG15-protein conjugation | 6 | 4.07E-05 | 1.65E-02 |
| GO:0019058 | Viral life cycle | 285 | 1.01E-04 | 3.96E-02 |
| GO:0035457 | Cellular response to interferon-alpha | 10 | 1.21E-04 | 4.64E-02 |

mRNA families. Utilizing Reactome [15, 16], six pathways were identified as significantly over-represented (FDR < 0.05; Table 2). These identified biological pathways are involved primarily in type I interferon signaling and antiviral mechanisms, represented predominantly by higher IFIT, ISG, HERC, BST, and OAS mRNA family expression. Similar to the biological processes, these pathways are largely represented by genes higher in expression in DEAD cattle. The predominant disease phenotypes identified by GLAD4U [17] consisted of viral-induced diseases, which were heavily influenced by certain genes increased in expression in DEAD cattle, including BST2, HERC5, IFIT1, ISG15, MX2, and OAS2 (Table 3). In summary, these analyses of biological processes, pathways, and disease phenotypes represented by DEG information indicate that cattle within the DEAD cohort have increased expression of genes involved in type I interferon production and viral-associated responses at arrival.

**Table 2. Top enriched signaling pathways (FDR < 0.05).**

| Gene Set | Description | Size | P Value | FDR |
|---|---|---|---|---|
| R-HSA-909733 | Interferon alpha/beta signaling | 69 | 0 | 0 |
| R-HSA-913531 | Interferon Signaling | 197 | 4.89E-15 | 4.87E-12 |
| R-HSA-1280215 | Cytokine Signaling in Immune system | 688 | 1.77E-10 | 1.18E-07 |
| R-HSA-168256 | Immune System | 1997 | 2.44E-08 | 1.22E-05 |
| R-HSA-1169410 | Antiviral mechanism by IFN-stimulated genes | 78 | 2.08E-07 | 8.28E-05 |
| R-HSA-1169408 | ISG15 antiviral mechanism | 71 | 6.75E-06 | 2.24E-03 |

## Protein-protein interactions and co-expression of DEGs

The 69 DEGs identified between DEAD and ALIVE were used to predict protein-protein interactions and gene product co-expression in STRING v11.0 (Fig 3) [18]. This analysis identified interactions consisting of 16 DEGs (nodes) connected by 57 interactions (edges), in which 13 of 16 DEGs were increased in expression in DEAD (Fig 3A). A strong pattern of co-

**Table 3. Top enriched disease phenotypes (FDR < 0.05).**

| Gene Set | Description | Size | P Value | FDR | Genes |
|---|---|---|---|---|---|
| PA444621 | Influenza, Human | 144 | 2.07E-11 | 4.56E-08 | BST2, HERC5, IFIT1, ISG15, ISG20, MX2, RSAD2, TIMD4 |
| PA444614 | Infection | 643 | 3.08E-11 | 4.56E-08 | APOBEC3A, BST2, HERC5, IFIT1, IFIT2, IFIT3, ISG15, ISG20, MX2, OAS2, RSAD2, TIMD4 |
| PA446038 | Virus Diseases | 580 | 2.15E-10 | 2.12E-07 | APOBEC3A, BST2, HERC5, IFIT1, IFIT2, IFIT3, ISG15, ISG20, MX2, OAS2, RSAD2 |
| PA444435 | Hepatitis | 253 | 1.85E-09 | 1.37E-06 | APOBEC3A, BST2, IFIT1, ISG15, ISG20, OAS2, RSAD2, TIMD4 |
| PA166170066 | Virological response | 282 | 4.34E-09 | 2.57E-06 | BST2, IFIT1, IFIT2, IFIT3, ISG15, ISG20, MX2, RSAD2 |
| PA447230 | HIV | 862 | 1.37E-08 | 6.73E-06 | APOBEC3A, BST2, HERC5, HNRNPA2B1, IFIT2, IFIT3, ISG15, ISG20, MX2, OAS2, RSAD2 |
| PA445746 | Stomatitis | 126 | 2.20E-08 | 9.31E-06 | BST2, IFIT1, IFIT2, IFIT3, MX2, RSAD2 |
| PA444020 | Encephalitis, Tick-Borne | 26 | 5.00E-08 | 1.85E-05 | IFIT1, IFIT2, OAS2, RSAD2 |
| PA444014 | Encephalitis | 89 | 7.68E-06 | 2.31E-03 | IFIT2, MX2, OAS2, RSAD2 |
| PA444445 | Hepatitis C | 195 | 7.82E-06 | 2.31E-03 | BST2, IFIT1, ISG15, OAS2, RSAD2 |
| PA445546 | Retroviridae Infections | 494 | 6.10E-05 | 1.29E-02 | APOBEC3A, BST2, HERC5, ISG15, MX2, RSAD2 |
| PA446213 | HIV Infections | 495 | 6.17E-05 | 1.29E-02 | APOBEC3A, BST2, HERC5, ISG15, MX2, RSAD2 |
| PA445640 | Sexually Transmitted Diseases | 496 | 6.24E-05 | 1.29E-02 | APOBEC3A, BST2, HERC5, ISG15, MX2, RSAD2 |
| PA446295 | Lentivirus Infections | 498 | 6.38E-05 | 1.29E-02 | APOBEC3A, BST2, HERC5, ISG15, MX2, RSAD2 |
| PA444601 | Immunologic Deficiency Syndromes | 500 | 6.52E-05 | 1.29E-02 | APOBEC3A, BST2, HERC5, ISG15, MX2, RSAD2 |
| PA446768 | Encephalitis, Viral | 66 | 1.14E-04 | 2.12E-02 | IFIT2, OAS2, RSAD2 |
| PA443855 | Dengue | 76 | 1.74E-04 | 3.03E-02 | IFIT3, OAS2, RSAD2 |

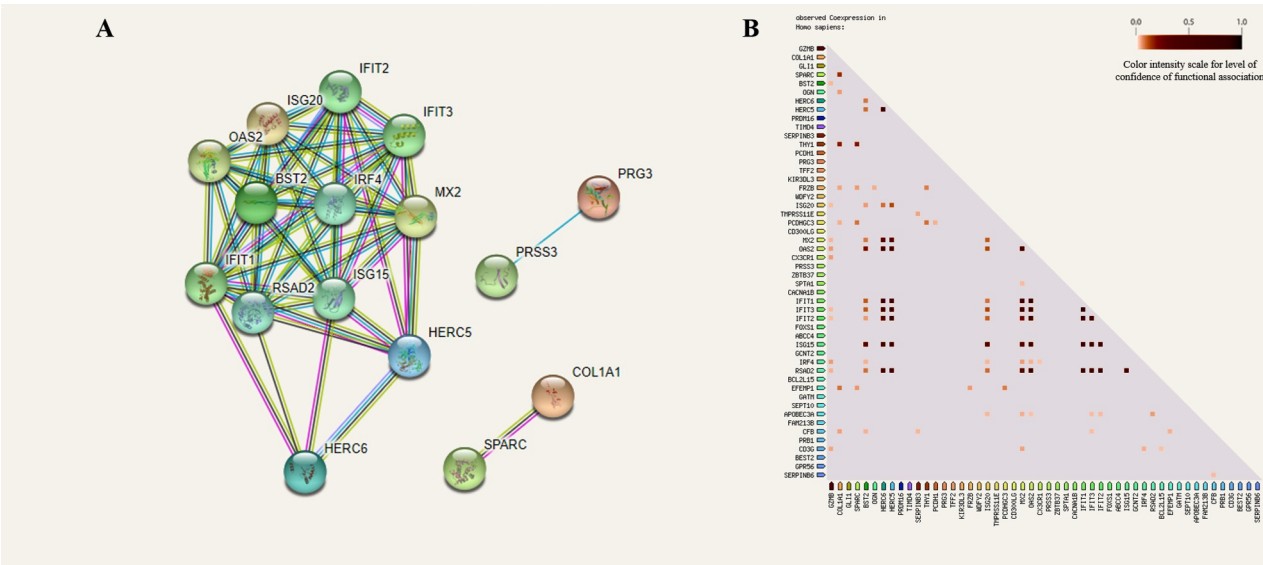

**Fig 3. Protein-protein interaction networking and co-expression analysis demonstrates close expressional patterns in DEGs.** A) Protein-protein interaction (PPI) analysis depicts strong interactions between multiple DEGs that are involved in type I interferon production and antiviral defense. These antiviral DEGs were all higher in expression in the DEAD cohort. All filled nodes represent DEGs with known or predicted three dimensional structures. Colored lines (edges) represent known interactions from curated databases (light blue) and published experimentation (pink) and predicted interactions from gene neighborhood/clusters (green), gene fusions (red), gene co-occurrences (dark blue), text mining (yellow), and co-expression (black). B) Gene co-expression analysis depicted in the triangle matrix demonstrates correlated expression patterns between individual gene products. The scale, from white/light red to dark red, indicates the level of confidence between each evaluated interaction.

expression between those 16 gene products was also identified as depicted in Fig 3B. The highest co-expression scores (> 0.75) were identified for IFIT1-3/5, SPARC, COL1A1, RSAD2, ISG15, HERC5/6, OAS2, and MX2. In summary, 12 DEGs that were increased at arrival in cattle who died from BRD are known to code for proteins that possess strong interactions and co-expression. Additionally, two genes increased in expression in ALIVE (COL1A1, SPARC) demonstrated strong predicted co-expression and interaction. All co-expression interactions and associated scores may be found in S2 File.

### *De novo* assembly and analysis of unmapped reads

From 6,968,239 unmapped reads contributed from all 6 samples, 6,953,629 survived quality trimming (99.79%) and were used to assemble a *de novo* transcriptome. The resulting 65,516 constructed contigs (S3 File) were analyzed against the NCBI non-redundant nucleotide (nt) database. Over 90% of the assembled contigs mapped to the *Bovidae* family, followed by alignments to various mammalian, bacterial (many part of the bovine microbiota), parasitic (*Onchocerca ochengi*), and fungal species (*Basidiomycota*) (Figs 4 and 5). Notably, the *de novo* assembly failed to map to any viral contigs within the NCBI nt database. Homology to viral DNA was not identified.

### Alignment of unmapped reads to viral genome sequences

Trimmed unmapped reads from each calf were analyzed in two parts: 1) against all known virus sequences and 2) against all known bovine viral pathogen sequences. Top hits from alignments against all viral assemblies demonstrated a sparse number of reads across all 6 animals (231–535 reads; total: 2,257, average: 376.2; S4 File). The majority of reads aligned to non-mammalian viruses, namely the *Choristoneura fumiferana* granulovirus and *Diolcogaster*

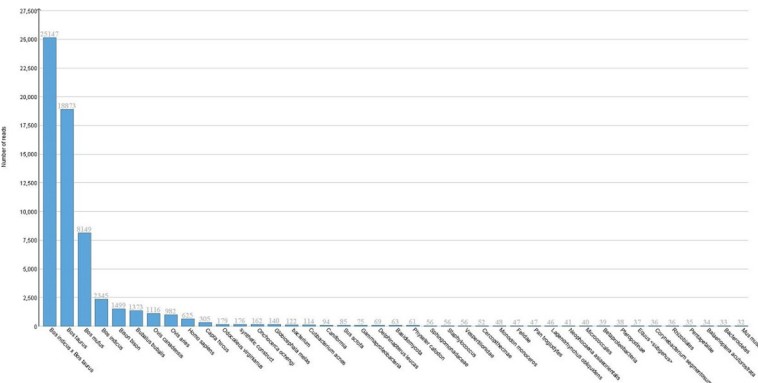

**Fig 4. Taxonomic identification of *de novo* constructed contigs.** Taxonomic distribution of *de novo* contig alignment against the NCBI nucleotide database, using only the top hit. The majority of hits were to *Bovidae* family assemblies and various mammalian species. Notably, no viral alignments were seen with the *de novo* contigs. The non-mammalian hits primarily consisted of bovine microbiota organisms, with few exceptions (*Basidiomycota*, *Onchocerca ochengi*). The total number of hits from each alignment are organized in descending order, from left to right.

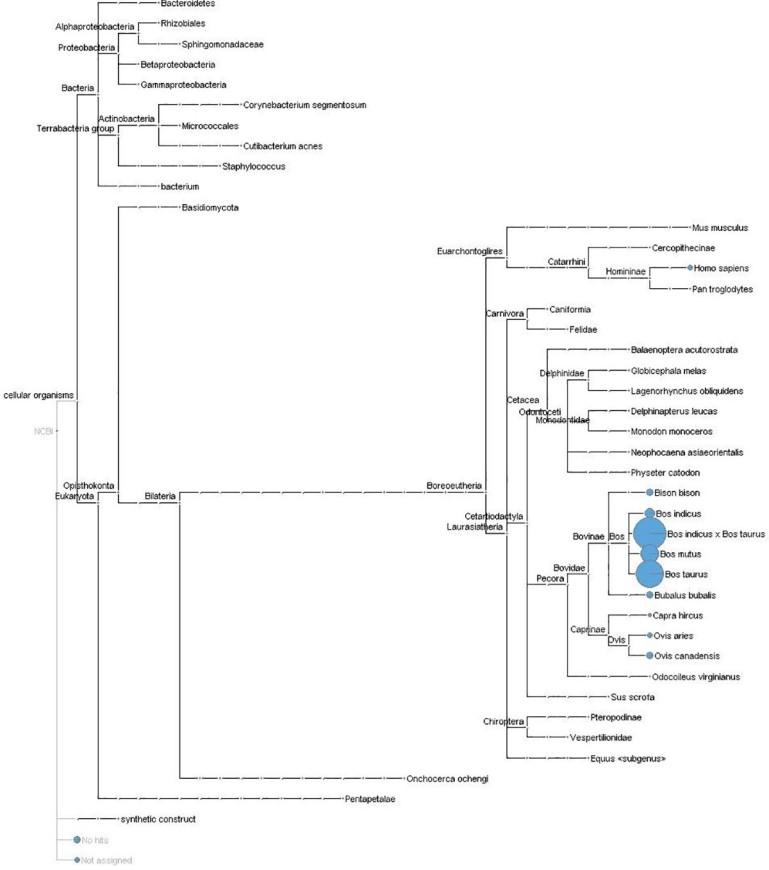

**Fig 5. Phylogenetic map of *de novo* constructed contigs.** Phylogenetic mapping of *de novo* assembled contigs with associated number of top hit alignments for each organism. The majority of alignments were seen with non-ARS-UCD1.2 *Bos taurus*, *Bos indicus*, and various ungulate assemblies. Assigned alignments are ordered within the phylogenetic tree, with a representative circle (in blue) based on the percentage of hits.

*facetosa* bracovirus, in addition to BeAn 58058 virus. However, independent analysis of these reads to the NCBI nt database indicated that these reads aligned to well conserved mammalian genes, such as U6 spliceosomal ncRNA. Due to the homologous nature of these reads to mammalian genes, any hits were considered irrelevant. Using the top hits from the alignments against known bovine viruses, all 6 animals possessed a relatively small number of reads (5082–7330 reads; total: 35,432, average: 5905.3; S5 File) that only aligned to BVDV1. These reads were extracted and realigned to both the NCBI nt database and the complete BVDV1 genome (NC_001461.1). When realigned to the NCBI nt database, the extracted reads aligned to the BVDV1 sequences U86599.1 (Pestivirus type 1 cytopathic genomic RNA, complete genome) and L13783.1 (Bovine viral diarrhea virus p125 protein gene, partial CDS). However, the top hits were consistently to ubiquitin C (UBC) mRNA sequences within *Bovidae* assemblies. When realigned to the NC_001461.1 complete BVDV1 genome, no alignment hits were detected.

## Discussion

This study builds upon our previous analysis of transcriptomes at arrival that were derived from post-weaned beef cattle that ultimately developed BRD versus cattle that remained healthy [14]. The present investigation was conducted with the intent to identify potential at-arrival biomarkers and pathways that indicate risk of BRD-associated mortality in post-weaned beef cattle. Our overarching goal with these studies is to identify gene expression profiles and biological pathways in blood samples at arrival that segregate with later BRD morbidity or mortality. By analyzing the transcriptomes of post-weaned cattle before they exhibit clinical signs of BRD, our approach will also improve understanding of the mechanistic basis of both susceptibility and resistance to BRD in this cohort. Previous research to determine early antemortem indications of BRD and risk of severity has yielded varied results [19–22]. At present, the diagnosis and classification of BRD is primarily assessed using clinical factors that have proven to be imprecise, effectively limiting BRD management [6, 10, 23]. Nonetheless, cattle diagnosed with BRD in this investigation using these same clinical factors (including DART scoring, treatment records, and average daily weight gain) exhibited differences in gene product and molecular pathway expression at arrival that ultimately segregated with their BRD-associated mortality [14].

In DEAD cattle, the expression of genes related to type I interferon production/signaling and viral defense were increased. These viral defense genes included IFIT1/2/3, IRF4, HERC5/6, OAS2, MX2, and ISG15/20. Several investigations have similarly identified these genes as relevant in cattle with prolonged inflammation and ongoing viral infection [24–27]. These findings, when considered with the increased expression of viral defense genes at arrival in cattle that ultimately died of BRD, suggest that cattle in the DEAD cohort were combating a viral agent at arrival. Though cattle that died of BRD in this investigation did not show clinical evidence of BRD at arrival, viral BRD is often subclinical and may initially present as an upper airway disease. Subclinical viral respiratory infection at arrival would not only account for the observed viral defense pathways in the cattle that died of BRD, but would facilitate secondary infectious processes in the lung that contribute to the observed BRD mortality [5, 25, 28, 29]. One challenge with our study is that differences in gene expression were characterized in peripheral blood. It has been suggested that the blood transcriptome represents an amalgamation of gene expression patterns and pathways in distinct physiological sites, such as airway epithelium, lymph nodes, and splenic tissue [30, 31]. Necropsy demonstrated that disease was limited to the lung in cattle that died. Overt disease was not evident at arrival, but subclinical disease cannot be ruled out. As these DEGs may indicate viral-induced disease at arrival,

focused investigation of these genes is warranted for larger future studies. The ability to delineate underlying subclinical viral disease mechanisms in beef cattle at arrival could allow for the development precision management techniques to more effectively and specifically treat or prevent disease, leading to more precise antimicrobial usage and decreased dissemination of antimicrobial resistance.

In this study, we identified increased expression of gene products that interact with toll-like receptor 4 (TLR4) and interleukin 6 (IL6) in both live and dead cohorts. While there was no difference in the expression of these specific genes between DEAD vs ALIVE cattle at arrival, we have previously described increased BGN, MARCO and POMC expression (known to be involved in TLR4-dependent pro-inflammatory pathways) in arrival blood transcriptomes from cattle that ultimately developed BRD when compared to cattle that remained clinically healthy [14]. IL6 and several other type I interferon-associated genes have been reported to be differentially expressed within lymph node samples of virus-challenged cattle [25, 26]. TLR4 possesses high avidity for lipopolysaccharide (LPS) and some viral structural proteins, and is capable of inducing type I interferons production and increased levels of IL6 [32–35]. Additionally, elevated levels of IL6 may reciprocatively induce type I interferon production, enhancing natural killer cell cytotoxic activity, M1 macrophage maturation, and interleukin 12 (IL12) production [36–39]. It is important to note that IL6 and TLR4 are not differentially expressed between DEAD and ALIVE groups but are predicted to be active based on associations with DEG products increased within each cohort. It is possible that TLR4 and IL6, relatively non-specific markers of inflammation, are initiated in both DEAD and ALIVE groups albeit through differing mechanisms. Several studies have demonstrated that TLR4 expression is increased in active respiratory disease and is responsible for proinflammatory cytokine production, in both viral and bacterial induced infections [40–43]. Furthermore, in ALIVE cattle, the increased expression of several proinflammatory genes was identified: CD300LG, COL1A1, CX3CR1, KIR2DL5A, LOC104968634 (NK2B), OGN, LOC782922 (PRXL2B), TARP. These gene products, largely involved in natural killer cell activation, leukocyte adhesion, prostaglandin synthesis, and initiation of the acquired immune system, possess known interactions or promotion of TLR4 and IL6 activity. In conjunction with TLR4 interactions, it is possible that the ALIVE cattle were actively combating extracellular antigens or etiological agents. The commonality between ALIVE and DEAD cattle is antigenic and immunogenic signaling without inflammatory mitigation. Notably, our research did not ascertain the order of TLR4 and IL6 association, therefore further research is necessary to define mechanistic characteristics and signaling order.

Some of the limitations of this study are the lack of antemortem pathogen identification, particularly viral isolation at arrival, and the relatively small number of biological replicates within each cohort. Due to technical and fiscal challenges associated with high-throughput sequencing, obtaining the optimal number of biological replicates remains a controversial aspect of RNA-Seq experimentation. However, it is generally accepted that three clean biological replicates per group is the minimum sample size necessary for meaningful inferential analysis [44, 45]. Modeling genes that were differentially expressed between live and dead cohorts identified increased antiviral pathways in the DEAD cohort. Accordingly, we utilized *de novo* alignment and BLAST toolkits to mine unmapped reads for viral sequences that would account for the gene expression changes and pathways identified in our study. Reads that fail to map to the host reference assembly have been previously used to identify pathogens within RNA-Seq datasets [46–50]. The *de novo* assembly reads aligned predominantly to annotated ungulate sequences (Figs 4 and 5). This is an expected occurrence in which the unmapped reads representing gene products in the tested cattle were not identified with the *Bos taurus* ARS-UCD1.2 reference assembly. This is not an uncommon occurrence with reference

assemblies from non-model organisms that reflects errors in the assembly's structural annotation and has been otherwise reported in transcriptomic experiments using cattle [46, 50].

A notable finding in this investigation is the lack of alignments to pathogenic organisms associated with BRD. This finding was consistent in three instances 1) when assembled contigs of the *de novo* transcriptome were mapped to all sequences in the NCBI non-redundant nt database; 2) when unmapped reads were mapped against all known viral sequences; and 3) when unmapped reads were mapped against all known bovine viral pathogen sequences. One key limitation of this experiment is the poly-A tail dependent capture of reads for library preparation. It is possible that pathogen genes within each whole blood sample were never captured prior to sequencing. Therefore, while we were unable to identify pathogen genes in these instances, it does not rule out the possibility of these cattle harboring etiological agents at arrival. All biological replicates possessed reads that matched only to BVDV1 sequences from the NCBI nt database. However, these reads matched solely to ubiquitin C (UBC) mRNA within *Bovidae* assemblies and no alignment was detectable when realigned to the NC_001461.1 complete BVDV1 genome. This demonstrates that the unmapped reads failed to align to viruses related to BRD, but rather aligned to bovine genome sequences that have been incorporated into BVDV1. It has been shown that several BVDV1 sequences possess *Bovidae* genomic sequence contamination, specifically to UBC mRNA. This finding agrees with the alignment discovery reported by Usman and colleagues [46]. Despite the absence of viral sequences, the identified DEGs and pathways provide evidence that the anti-viral mechanisms were activated at arrival in cattle within the DEAD cohort.

## Conclusions

This study explored the at-arrival whole blood transcriptomes and differentially expressed genes of diseased cattle, identifying significant gene products and pathways that differentiate cattle that die from naturally acquired respiratory disease and those that develop BRD but survive. Our results demonstrate that cattle developing clinical BRD possess increased expression of genes involved in proinflammation and immune responses at arrival and share TLR4 and IL6 activity. Cattle that died from BRD demonstrated increased type I interferon and antiviral-associated gene product expression at arrival. Although the unmapped reads did not align to viral genomes, our at-arrival findings highlight candidate gene expression profiles that herald viral respiratory infections, prior to the identification of overt BRD. These findings may support the development of predictive viral disease assays and thus warrant further investigations comparing current pathogen detection techniques with the identification of these candidate biomarkers.

## Materials and methods

### Study design

All animal use and procedures were approved by the Mississippi State University Institutional Animal Care and Use Committee (IACUC protocol #17–120). This study examined whole blood transcriptomes at arrival from crossbred bulls (n = 5) and a steer (n = 1) that went on to develop clinical BRD within 28 days following facility arrival. These six animals were categorized into two groups based on BRD-attributed mortality. Group 1 (DEAD; n = 3) cattle died of naturally occurring BRD despite antimicrobial and supportive treatment; all animals within the DEAD cohort succumbed to BRD prior to administrated euthanasia. Group 2 (ALIVE; n = 3) cattle were treated for naturally occurring BRD, but subsequently recovered after one or more therapeutic courses of treatment. Animals in this investigation were a subset of a randomized experiment pertaining to the effect of vaccination and deworming on post-weaned

beef cattle health and growth factors [51]. Animals enrolled in this study were purchased from local commercial livestock auctions within Mississippi and housed at the H. H. Leveck Animal Research Center at Mississippi State University. Further information involving animal management and enrollment selection is addressed in S6 File and in detail in our previous studies [14, 51]. At arrival, jugular blood samples were collected into Tempus Blood RNA tubes (Applied Biosystems), and then frozen and stored at -80˚ C until analysis. RNA extraction, quality assessment, cDNA library preparation, and RNA sequencing (80 million reads/sample) was performed by the UCLA Technology Center for Genomics and Bioinformatics (UCLA TCGB, Los Angeles, CA, USA) as previously described [14]. The whole blood transcriptomes of the six individuals examined in this investigation have been previously contrasted against at arrival whole blood transcriptomes from cattle that failed to develop clinical signs of BRD [14]. RNA sequence reads are available in the NCBI sequence read archive (https://www.ncbi.nlm. nih.gov/geo/query/acc.cgi?acc=GSE136176). In contrast to our prior work, this investigation contrasts gene expression at arrival in cattle that went on to develop more clinically severe BRD versus less clinically severe BRD.

## Differential gene expression analysis

Program parameters and alignment statistics for reference-based gene count matrix construction were as previously described [14]. All sequence alignment map (SAM) files produced by HISAT2 were converted to binary alignment map (BAM) files, sorted, and indexed with SAMtools v1.9 [52]. All unmapped reads were extracted using SAMtools for further exploration. Reference-guided assembly and assessment were performed with StringTie v2.0 and GffCompare v0.11.4, respectively [53–55]. Gene-level read counts for each sample were calculated in Python v2.7.17 with the program prepDE.py [56].

Filtering, normalization, and analysis of gene counts was performed in R, utilizing the Bioconductor [57] software package edgeR v3.26.8 [58, 59]. Data for all six biological replicates were categorized into two groups based on BRD-associated mortality (n = 3 DEAD; n = 3 ALIVE). Filtering of low gene counts was performed as described by Chen and colleagues using a total count per million minimum of 0.2 across at least three samples [60]. Library sizes were normalized using the trimmed mean of M-values method [61] in edgeR. Unsupervised clustering of the aligned reads was performed using multidimensional scaling (MDS) in order to plot differences in expression profiles between the 6 animals [62]. Distances between samples on the MDS plot represent 'leading fold change', defined as the root-mean-square average of the log-fold-changes for the genes best distinguishing each pair of samples. Differentially expressed genes (DEGs) were identified in edgeR using likelihood ratio testing (glmLRT function) to improve the ability to analyze samples with large gene count dispersions and low abundance counts [59]. Differential gene expression was considered significant with a false discovery rate (FDR) of $\leq$ 0.10 [63].

## Biological interpretation of gene expression data

A heat map of the DEGs was created using the R package pheatmap v1.0.12 [64]. Gene Ontology (GO) analysis, biological pathways, and disease associations of the DEGs identified between DEAD and ALIVE groups were analyzed with the WEB-based Gene SeT AnaLysis Toolkit (WebGestalt 2019; http://www.webgestalt.org/), using the human orthologs of all bovine DEGs [65]. Overrepresented biological pathway analysis was performed utilizing the pathway database Reactome [15, 16]. Disease association analysis with the list of DEGs was performed using the GLAD4U functional database [17]. Analysis parameters within WebGestalt 2019 included between 5 and 3000 genes per category and an FDR cutoff of < 0.05 for

significance. Protein-protein interaction networks and protein co-expression analysis was conducted with the Search Tool for the Retrieval of Interacting Genes (STRING; https://string-db.org/) database v11.0 [18] using human orthologs of the bovine DEGs. All interactions required a minimum interaction (confidence) score of 0.900, defined as the highest confidence score in STRING v11.0.

## Analysis of unmapped reads

All reads in this study were originally aligned to the *Bos taurus* reference assembly ARS-UCD1.2. The unmapped reads which failed to align were extracted using SAMtools view -b option. The subsequent BAM files were converted into unique paired end fastq files with BEDtools v2.26.0 bamtofastq option [66]. Unmapped fastq files were retrimmed and quality assessed with Trimmomatic v0.38 and FastQC v0.11.9, respectively, in order to eliminate the potential of poor quality or inadequate length of sequences leading to misalignment. Trimming parameters were: 1) leading and trailing bases of each read were removed if their base quality score was below 3, 2) each read was scanned with a 3-base pair sliding window, removing read segments below a minimum base quality score of 15, and 3) sequences below a read length of 40 bases were removed. Read samples were concatenated based on directionality, and then assembled *de novo* into contigs using Trinity v2.8.5 by employing the program's default protocols [67]. Unmapped read trimming and *de novo* alignment statistics are provided in S3 File.

Nucleotide sequence homology of the *de novo* transcriptome was explored against the NCBI non-redundant nucleotide database (ftp://ftp.ncbi.nlm.nih.gov/blast/db/; accessed September 30, 2019) with NCBI-blast v2.9.0+ (ftp://ftp.ncbi.nlm.nih.gov/blast/executables/blast+/LATEST/; blastn parameter, default settings) [68]. The top alignment hit for each contig was identified and used to create a blastn file that was further analyzed for phylogenetic grouping and characterization using MEGAN Community Edition v6.18.3 (https://github.com/danielhuson/megan-ce) using the program's default protocols for taxonomic identification [69, 70].

Due to likely inaccuracies in the *de novo* assembly that trace to false chimeric contigs [71, 72], and persistence of reads that were not incorporated into the *de novo* transcriptome, we therefore analyzed all trimmed reads against bovine viral pathogen sequences downloaded from NCBI (https://www.ncbi.nlm.nih.gov/labs/virus/vssi; accessed January 6, 2020). A total of 1657 nucleotide files were downloaded and utilized as subject sequences by selecting only complete nucleotide sequence types found from *Bos taurus* (taxid: 9913) hosts. In addition, we also aligned reads from cattle that were not incorporated into the *de novo* transcriptome to all known viral sequences at NCBI (release 200, https://ftp.ncbi.nlm.nih.gov/refseq/release/complete/). Local alignment was performed with the NCBI-blast v2.9.0+ blastn option, using the same settings parameters as previously mentioned. The resulting blastn file from each sample was explored with MEGAN Community Edition v6.18.3. Top hits were scrutinized for potential genomic DNA contamination in the reference subjects by re-aligning the respective cDNA of the sample read sequence to both the official genome assembly that it annotated as and to the NCBI nucleotide database.

## Supporting information

**S1 File. Complete list of differentially expressed genes and associated statistics between DEAD and ALIVE groups.**
(XLSX)

**S2 File. Gene names, annotations, and co-expression values of STRING interaction analysis.**
(XLSX)

**S3 File. Unmapped read trimming and de novo alignment statistics.**
(XLSX)

**S4 File. Nucleotide BLAST summary of unmapped reads against all viral assemblies.**
(PDF)

**S5 File. Nucleotide BLAST summary of unmapped reads against bovine viral assemblies.**
(PDF)

**S6 File. Descriptive statistics of animals within DEAD and ALIVE groups.**
(CSV)

## Acknowledgments

The authors would like to thank the students and staff of the Mississippi Agricultural and Forestry Experiment Station (MAFES), Mississippi State University College of Veterinary Medicine, and Mississippi State University Animal and Diary Science Department for their assistance in animal care and sample collection. Additionally, we would like to thank Merrilee Thoresen, Daniele Doyle, and Kathleen Barton for their technical assistance and insight throughout this experiment.

## Author Contributions

**Conceptualization:** Matthew A. Scott, Amelia R. Woolums, Cyprianna E. Swiderski, David R. Smith.

**Data curation:** Matthew A. Scott, Amelia R. Woolums, David R. Smith, Brandi B. Karisch.

**Formal analysis:** Matthew A. Scott, Cyprianna E. Swiderski, Andy D. Perkins, Bindu Nanduri.

**Investigation:** Matthew A. Scott, Amelia R. Woolums, Cyprianna E. Swiderski, David R. Smith, Brandi B. Karisch, John R. Blanton.

**Methodology:** Matthew A. Scott, Cyprianna E. Swiderski, Andy D. Perkins, Bindu Nanduri.

**Project administration:** Amelia R. Woolums, David R. Smith, Brandi B. Karisch, William B. Epperson, John R. Blanton.

**Supervision:** Amelia R. Woolums, Cyprianna E. Swiderski, Brandi B. Karisch, William B. Epperson, John R. Blanton.

**Validation:** Matthew A. Scott, Andy D. Perkins, Bindu Nanduri.

**Visualization:** Matthew A. Scott.

**Writing – original draft:** Matthew A. Scott.

**Writing – review & editing:** Matthew A. Scott, Amelia R. Woolums, Cyprianna E. Swiderski, Andy D. Perkins, Bindu Nanduri, David R. Smith, Brandi B. Karisch, William B. Epperson, John R. Blanton.

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
