## [Decision Letter · Decision Letter 0]

8 Jan 2021

PONE-D-20-27349

Comprehensive at-arrival transcriptomic analysis of post-weaned beef cattle uncovers type I interferon and antiviral mechanisms associated with bovine respiratory disease mortality

PLOS ONE

Dear Dr. Scott,

Thank you for submitting your manuscript to PLOS ONE. After careful consideration, we feel that it has merit but does not fully meet PLOS ONE’s publication criteria as it currently stands. Therefore, we invite you to submit a revised version of the manuscript that addresses the points raised during the review process.

We look forward to receiving your revised manuscript.

Kind regards,

Maria del Mar Ortega-Villaizan

Academic Editor

PLOS ONE

Additional Editor Comments:

The manuscript is of interest but minor corrections indicated by reviewers should be addressed before being accepted for publication.

Journal Requirements:

Reviewers' comments:

Reviewer's Responses to Questions

**Comments to the Author**

1. Is the manuscript technically sound, and do the data support the conclusions?

Reviewer #1: Yes

Reviewer #2: Partly

2. Has the statistical analysis been performed appropriately and rigorously? 

Reviewer #1: Yes

Reviewer #2: I Don't Know

3. Have the authors made all data underlying the findings in their manuscript fully available?

Reviewer #1: Yes

Reviewer #2: Yes

4. Is the manuscript presented in an intelligible fashion and written in standard English?

Reviewer #1: Yes

Reviewer #2: Yes

5. Review Comments to the Author

Reviewer #1: Thank you for the opportunity to review this manuscript. This study characterizes the whole blood transcriptome of cattle that survived and succumbed to the effects of BRD in an attempt to identify biomarkers that may be useful in determining risk classification. Overall, I find this article very interesting and quiet well written. I have a few comments for the authors to consider:

General comments:

While I understand the limitations of a project such as this, only 6 animals were evaluated in this study - 3 dead and 3 alive. How do the authors think that this small sample size might have impacted outcome? would the results be different if a larger number of animals were included? Could some comment regarding this be placed in the discussion?

Abstract

Lines 24-25: The statement beginning with "Clinical diagnosis" is somewhat confusing. While I understand the intent, visual detection methods are both non-specific and insensitive. It is stated more clearly in the introduction (Lines 60-61).

Discussion

The results of the study would suggest that animals that succumb to the effects of BRD are potentially combating viral pathogens at the time of facility arrival. How do the authors see the results of this study being used in the management of high-risk beef stocker calves to reduce antimicrobial use and improve survival?

what would a future direction for this work entail? In other words, how could this data be used in the future at a herd level?

Reviewer #2: This is an interesting manuscript and provides interesting information on which to build future studies. While this manuscript is concise, I think it would benefit from a slight re-write to make it less jargon-filled and more accessible to investigators with a broader range of expertise, especially since this is an area where a multidisciplinary approach is vital if we are going to be successful at tackling this problem.

Another criticism is that the conclusions seem to suggest that this technique/findings provide a method for early detection of BRD infection at arrival, however this was apparently the objective of the project, nor is this directly addressed by the findings.

More specific items are below:

Line 56: I'm not familiar with the term “newly placed,” perhaps recently transported?

Line 75: This sentence is not quite right. Too many “to”s and you need to compare something to something else.

Line 110: Does this mean that these 69 genes were differentially expressed between ALL the dead and all the alive, and any differences within each group were not included? For example, if DEAD 1 and 2 upregulated gene A, but not DEAD 3, was this gene included or not?

Line 164 (Figure 3): These figures are incredibly hard to read. The colors in figure A are difficult to differentiate, especially with the interconnected lines, and the reflection on the bubbles is distracting and complicates the image needlessly. In figure B it is difficult to see the lighter pink squares. I would also strongly recommend checking with guidelines for color blind friendly coloring in figures.

Line 243: This seems like a fairly strong statement without any actual evidence of infection?

Line 304: How often are cattle viremic for BRD pathogens? Would you expect to find viral transcripts if the animals are indeed infected at arrival?

Line 324: This conclusion seems to state that these candidate genes provide an early detection method for BRD, however this study did not address that question. Also, if this is the statement the authors wish to make, it would be useful to compare this with early pathogen detection techniques which would be cheaper and more practical at arrival.

6. PLOS authors have the option to publish the peer review history of their article (what does this mean?). If published, this will include your full peer review and any attached files.

Reviewer #1: **Yes: **Brent Credille, DVM, PhD, DACVIM

Associate Professor

Food Animal Health and Management Program

Department of Population Health

College of Veterinary Medicine

University of Georgia

Reviewer #2: **Yes: **Meera Heller

---

## [Author Response · Author response to Decision Letter 0]

22 Jan 2021

Please see the document titled "Response to Reviewers" for all comments and revisions.

---

## [Decision Letter · Decision Letter 1]

14 Apr 2021

Comprehensive at-arrival transcriptomic analysis of post-weaned beef cattle uncovers type I interferon and antiviral mechanisms associated with bovine respiratory disease mortality

PONE-D-20-27349R1

Dear Dr. Scott,

We’re pleased to inform you that your manuscript has been judged scientifically suitable for publication and will be formally accepted for publication once it meets all outstanding technical requirements.

Kind regards,

Maria del Mar Ortega-Villaizan

Academic Editor

PLOS ONE

Additional Editor Comments (optional):

Reviewers' comments:

Reviewer's Responses to Questions

**Comments to the Author**

1. If the authors have adequately addressed your comments raised in a previous round of review and you feel that this manuscript is now acceptable for publication, you may indicate that here to bypass the “Comments to the Author” section, enter your conflict of interest statement in the “Confidential to Editor” section, and submit your "Accept" recommendation.

Reviewer #1: All comments have been addressed

Reviewer #3: All comments have been addressed

2. Is the manuscript technically sound, and do the data support the conclusions?

Reviewer #1: Yes

Reviewer #3: Yes

3. Has the statistical analysis been performed appropriately and rigorously? 

Reviewer #1: Yes

Reviewer #3: Yes

4. Have the authors made all data underlying the findings in their manuscript fully available?

Reviewer #1: Yes

Reviewer #3: Yes

5. Is the manuscript presented in an intelligible fashion and written in standard English?

Reviewer #1: Yes

Reviewer #3: Yes

6. Review Comments to the Author

Reviewer #1: The authors adequately addressed my comments. Thank you for the opportunity to review this manuscript.

Reviewer #3: The reviewer comments have been adequately addressed. This will be an excellent addition to the literature on bovine respiratory disease.

7. PLOS authors have the option to publish the peer review history of their article (what does this mean?). If published, this will include your full peer review and any attached files.

Reviewer #1: **Yes: **Brent Credille, DVM, PhD, DACVIM

Reviewer #3: No

---

## [Editor Report · Acceptance letter]

16 Apr 2021

PONE-D-20-27349R1 

Comprehensive at-arrival transcriptomic analysis of post-weaned beef cattle uncovers type I interferon and antiviral mechanisms associated with bovine respiratory disease mortality 

Dear Dr. Scott:

I'm pleased to inform you that your manuscript has been deemed suitable for publication in PLOS ONE. Congratulations! Your manuscript is now with our production department. 

Kind regards, 

on behalf of

Dr. Maria del Mar Ortega-Villaizan 

Academic Editor

PLOS ONE